# Gut commensal bacteria influence colorectal cancer development by modulating immune response in AOM/DSS-treated mice

Danlei Zhou,[1,2] Yujing Sun,[1,2] Peipei Ding,[1,2] Xiaochao Wang,[1,2] Ling Li,[1,2] Luying Li,[1,2] Xinyue Lv,[1,2] Tian Liao,[3,4] Jianfeng Chen,[1,2] Wei Zhang,[1,2] Qi Wang,[1,2] Qing-Hai Ji,[3,4] Feng Gao,[5,6,7] Weiguo Hu[1,2,8]

**ABSTRACT** The gut microbiota has been closely associated with the pathogenesis of colorectal cancer (CRC). However, precise identification of particular microorganisms promoting CRC carcinogenesis, and more importantly those blocking tumor development, has been challenging based on human gut microbiota profiling studies. With a well-established azoxymethane/dextran sodium sulfate induction murine CRC model, we found a subset of mice consistently failed to develop CRC. This genetically homogeneous but cancer-refractory population gave us a unique opportunity to reveal that the microbial compositions between mice with and without CRC formation are indeed distinct, indicating key different gut microbiota between those groups are responsible for the differential susceptibility of the animals to CRC development. Our analysis revealed that *Ruminococcus flavefaciens* (R.f) and *Fibrobacter succinogenes* (F.s) were significantly enriched in CRC-free mice, while the presence of *Eubacterium dolichum* (E.d) was dramatically reduced. The correlative evidence was further substantiated as important causal factors, with subsequent bacteria intragastric administration experiments demonstrating independent, protective roles of R.f and F.s and a correspondingly detrimental role of E.d in inflammation-induced CRC initiation. Notably, E.d strongly activates NF-κB and promotes the local accumulation of myeloid-derived suppressor cells and macrophages. Significant disturbance of gut immune homeostasis, therefore, might be a critical trigger leading to subsequent CRC development. These findings indicate a clear direction for precise and rational gut microbiota-mediated CRC prevention.

**IMPORTANCE** There is a complex ecosystem of different microbes residing within the gut, which is highly relevant to health and diseases. The causal linkage between specific gut microbes and the development of colorectal cancer has been established with a mouse model, pinpointing specific bacteria species either promoting or preventing colorectal cancer development. A key aspect of these gut residual bacteria in colorectal cancer development is through exaggerating or easing gut inflammation. Therefore, by taking probiotics composed of corresponding cancer-preventing bacteria from human microbiota, it can be an effective and economic way to reduce human colorectal cancer risks.

**KEYWORDS** AOM/DSS, microbiota, colorectal cancer, *Eubacterium dolichum*, NF-κB

Colorectal cancer (CRC) is projected to be responsible for the third most prevalent new cancer cases and related fatality worldwide (1). Since the mid-1990s, the incidence of CRC has been rising at a particularly alarming rate, especially in the developed world. While the specific causes for this trend remain elusive, it is speculated

**Peer Reviewer** Huoxiang Zhou, Henan Institute of Medical and Pharmaceutical Science, The Fifth Affiliated Hospital of Zhengzhou University, Zhengzhou, China

Address correspondence to Weiguo Hu, weiguohu@fudan.edu.cn.

The authors declare no conflict of interest.

See the funding table on p. 17.

that the sedentary lifestyle and unhealthy diet adopted by recent generations might be critical factors (1). On the contrary, inflammatory bowel disease (IBD), which includes Crohn's disease and ulcerative colitis, is also skyrocketing during the same period. The intimate association between IBD and an increased risk of CRC is widely accepted (2), indicating lifestyle and dietary factors related to gut chronic inflammation can be pivotal in CRC development. While the risk of CRC is directly proportional to the duration and extent of inflammation, with a cumulative cancer incidence reaching as high as 30% in patients with longstanding UC with widespread colonic involvement (3), a vast majority of IBD patients never develop CRC. This discrepancy suggests that even in the context of chronic gut inflammation, there are potentially complex compounding factors influencing CRC development. It is challenging to devise a well-controlled study in IBD patients to pinpoint potential causal factors facilitating or blocking CRC, as genetic susceptibility, inflammatory progression, lifestyle, and diet are dynamically complex and heterogeneous in the patient population. In addition, we are particularly interested in whether gut microbial community structure or resident bacteria can be responsible for disease progression toward CRC or potentially block tumorigenesis in the context of a chronically inflamed gut. Previous findings have shown that continuous alterations in the gut microbiota can directly contribute to tumor formation (4–9). Therefore, we hypothesize that if a close causal relationship can be established between key gut microbiota and CRC development, we can rationally design CRC prevention probiotics specifically for high-risk patients with refractory inflammatory bowel diseases.

Studies have shown significant changes in the microbial community structures of CRC patients, particularly in cases of inflammation (10, 11). There is accumulating evidence supporting the pivotal role of specific gut microbiota in either promoting CRC development or suppressing cancer. Metagenomic sequencing of stool samples has shown that *Bacteroides* is abundant in CRC patients, while *Ruminococcus* is more prevalent in healthy individuals (12). In two independent cohort studies, 12 fecal bacterial taxa were found to be abundant in CRC patients, including two *Porphyromonas* species inducing cellular senescence by secreting the bacterial metabolite butyrate (13). Feeding the feces of the CRC patients to mice promoted colorectal tumorigenesis in both germ-free and conventionally raised mice predisposed with genotoxic damage, indicating the causal contribution of CRC patient-associated microbiota to tumorigenesis (14). Conversely, when distinct human-like gut microbiota structures were predisposed into germ-free mice, and then chemically induced CRC, the number of tumors formed in genetically homogenous mice strongly correlated with their distinctive microbiome structure (15), a notion supported by similar studies on the protective role of commensal bacteria in limiting epithelial overproliferation (16). Similar studies have shown the protective role of commensal bacteria in limiting epithelial overproliferation and the significant reduction in tumor size and numbers by treating mice with antibiotics (17). Therefore, the risk of CRC is clearly related to an imbalance with gut microbiota composition and its associated metabolites. While the analysis of microbiota structure in feces provides appealing predictive power in forecasting the risk of CRC (18), the development of rational and effective microbe-based cancer preventive agents with significant influence on CRC disease outcome is still in its infancy due to the limited identification and rigorous validation of functional gut flora.

Modeling colitis-associated colorectal cancer in mice has gained popularity for its reproducibility, potency, low price, and ease of use. Mice pre-treated with genotoxic agents, such as azoxymethane (AOM), are particularly susceptible to rapid CRC-like tumor development through mimicking a chronic gut inflammation condition via repeated cycles of administration of dextran sodium sulfate (DSS) through drinking water (causing cycles of significant colonic epithelium damage and repair) (19). The established AOM/DSS model has been widely used and reported a close to 100% incidence of multiple CRC-like colonic adenocarcinoma per treated mice within 7–10 weeks of DSS treatment (19–21). Given the power of this widely adopted model, we speculate that the small minority of tumor-free mice after the AOM/DSS treatment might

reflect a particularly strong cancer prevention potential of their unique gut microbiota, given their identical genetic background, diet, and treatment scheme. A closer comparative examination using a fecal microbiota-centered approach enables us to identify key bacteria as particularly potent blockers for CRC development within the context of a chronically inflamed gut.

## MATERIALS AND METHODS

### Bacterial culture and oral infection

Bacterial strains *Eubacterium dolichum* (ATCC29143), *Ruminococcus flavefaciens* (ATCC49949), and *Fibrobacter succinogenes* (ATCC51214) were obtained from American Type Culture Collection (ATCC). The strains were cultivated under anaerobic conditions at 37°C using brain heart infusion (BHI) culture medium with oxygen-free mixed gas (95% $N_2$ and 5% $CO_2$) in the anaerobic incubator (DG250, Don Whitley Scientific). Each test tube containing the medium was closed tightly with an inner butyl rubber stopper equipped with an outer screw cap. Inoculation of pre-cultivated cells to the media was carried out under the stream of oxygen-free mixed gas (95% $N_2$ and 5% $CO_2$). The bacteria were passed every 2 days, and the mice began to be fed with the 4th to 5th generations of bacteria.

Prior to the implantation of specific bacteria, standard specific pathogen-free (SPF) reared mice underwent gut microbiota depletion through the administration of an antibiotics cocktail. The treatment mice were provided with drinking water containing streptomycin (1.0 g/L, Solarbio), neomycin (1.0 g/L, Solarbio), ampicillin (1.0 g/L, Solarbio), gentamicin (1.0 g/L, Solarbio), metronidazole (1.0 g/L, Solarbio), and vancomycin (0.5 g/L, Solarbio) for 1 week before AOM injection. The bacteria were cultured at 37°C under anaerobic conditions and administered to the mice 1 week after AOM injection (day 15) on alternate days during each round of normal water feeding (Fig. 1A) until the animals were sacrificed. For oral infection, mice were orally gavaged with 0.2 mL of live bacterial suspension with $OD_{600\,nm}$ readings between 1.0 and 1.5 ($1 \times 10^9$ CFU/mL).

### Animals and treatment

Male C57BL/6J mice aged 4–5 weeks (SLAC Laboratory Animal Co., Ltd.) were used for the experiments and housed under SPF conditions at the animal facility of the Shanghai Medical School, Fudan University, China. CRC was induced following standard protocols with minor modifications (20). Briefly, on days 1 and 8, the mice (about 7–8 weeks old) were intraperitoneally injected twice with AOM (9 mg/kg body weight; Sigma, A5486) and maintained on a regular diet and water for 7 days. On day 15, mice were switched to water containing 1.5% DSS (MP Biomedicals, Santa Ana) for 1 week, followed by regular water for 2 weeks. After three full colon damage/recovery cycles (1 week with DSS, 2 weeks without), on day 78, the mice were euthanized. For experiments that require bacterial gavage, mice were injected with AOM (day 1) after 1 week antibiotics cocktail treatment, followed by 1 week recovery period. The bacteria were administered to mice on alternate days during each round of normal water feeding. Normal control mice were administered with 0.2 mL BHI medium. Mice in the experimental group were intragastrically administered with 0.2 mL live bacteria fresh from the anaerobic incubator (within 2 h). The full AOM/DSS CRC modeling process is illustrated in Fig. 1A. The study period may be adjusted according to the actual tumor formation. In the experiment with R.f treatment (Fig. 2A), four colon damage/recovery cycles were employed to allow more time for tumor development. For the experiment with E.d treatment (Fig. 3A), the last cycle of water feeding was extended for an additional week before euthanization.

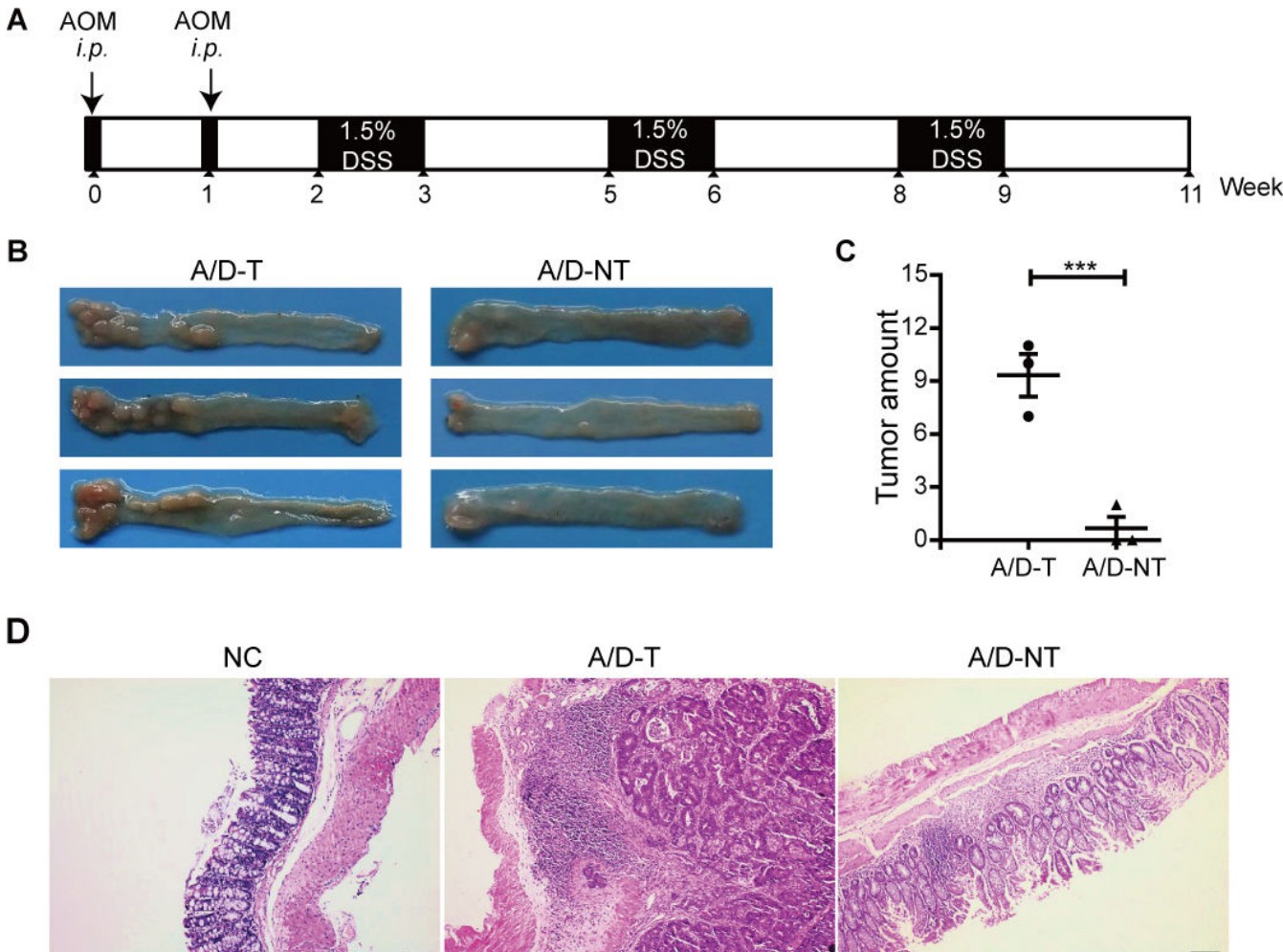

**FIG 1** AOM/DSS stimulated colonic inflammation and colon tumor formation in mice. (A) Mice were treated with AOM and 3 cycles of 1.5% DSS. (B) Representative images of lumps in the distal colons of mice treated with AOM/DSS. A/D-T: mice with tumors after AOM/DSS; A/D-NT: mice without tumors after AOM/DSS. (C) Numbers of lumps observed at the end of the experiment when mice were colonized. (D) Histologic examination with hematoxylin and eosin staining was performed (original magnification ×100, scale bar = 100 µm). Data are expressed as the mean ± standard error of the mean. $n = 3$. ***$P < 0.001$.

## Histological analysis

For histological analysis, 1–2 cm colon segments from control and AOM/DSS-treated mice were fixed overnight in 10% formalin, embedded in paraffin blocks, cut into 5 µm sections, and stained with hematoxylin and eosin as previously described (22).

## 16S rRNA gene amplification and sequencing

Fecal samples were regularly collected from all mice from metabolic cages until euthanization. After euthanization of the mice on day 78, three mice were found to have not developed visually apparent tumors upon biopsy. Their corresponding fecal samples prior to euthanization were identified. Similarly, corresponding control fecal samples were taken from three representative mice with typical colon tumor burden (Fig. 1B). Total bacterial DNA was then extracted using a hexadecyltrimethylammonium bromide/sodium dodecyl sulfate method (23). DNA concentration and purity were determined via 1% agarose gels. DNA was diluted to 1 ng/µL using sterile water. PCR amplifications were conducted in triplicate with the primer sets 515F (5′-GTGCCAGCAGCCGCGGTAA-3′) and 806R (5′-GGACTACCAGGGTATCTAAT-3′), which amplify the V4 region of the bacterial 16S rRNA gene. All PCRs were carried out in 30 µL reactions with Phusion High-fidelity PCR

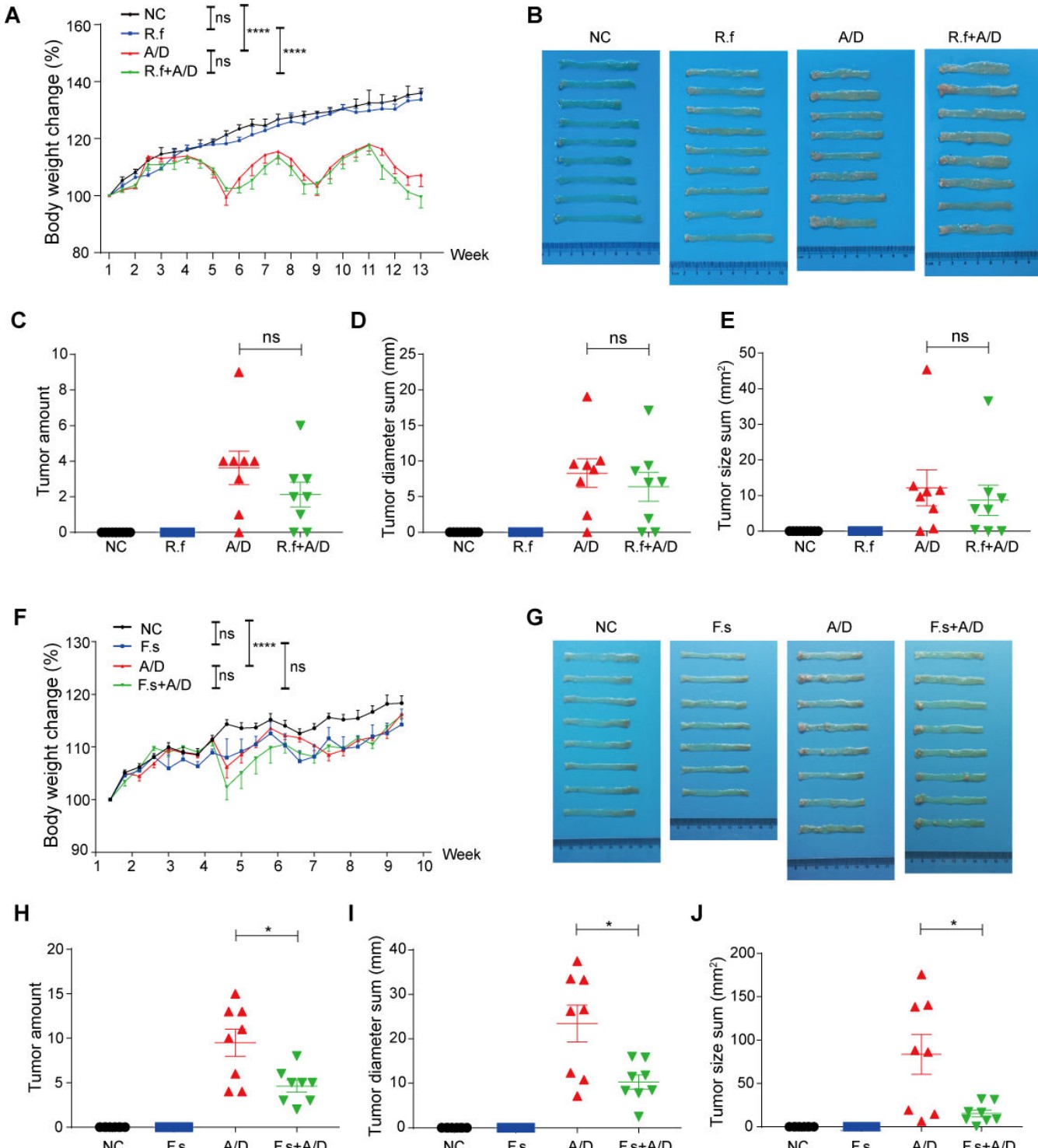

**FIG 2** R.f and F.s reduced colonic inflammation or colon tumor formation in mice treated with AOM/DSS. We orally infected mice with R.f and F.s between AOM administration and DSS-induced tissue inflammation and compared those with mice undergoing the AOM/DSS regimen without external bacterial implantation. (A) Body weight change in mice treated with AOM/DSS and 4 cycles of R.f. (B) Images of lumps in the distal colons of mice treated with AOM/DSS and 4 cycles of R.f. (C–E) Numbers (C), diameters (D), and sizes (E) of tumors at the end of the experiment when mice (treated with AOM/DSS and 4 cycles of R.f) were sacrificed. (F) Body weight change in mice treated with AOM/DSS and 3 cycles of F.s. (G) Images of lumps in the distal colons of mice treated with AOM/DSS and 3 cycles of F.s. (H–J) Numbers (H), diameters (I), and sizes (J) of tumors observed at the end of the model when mice (treated with AOM/DSS and 3 cycles of F.s) were sacrificed.

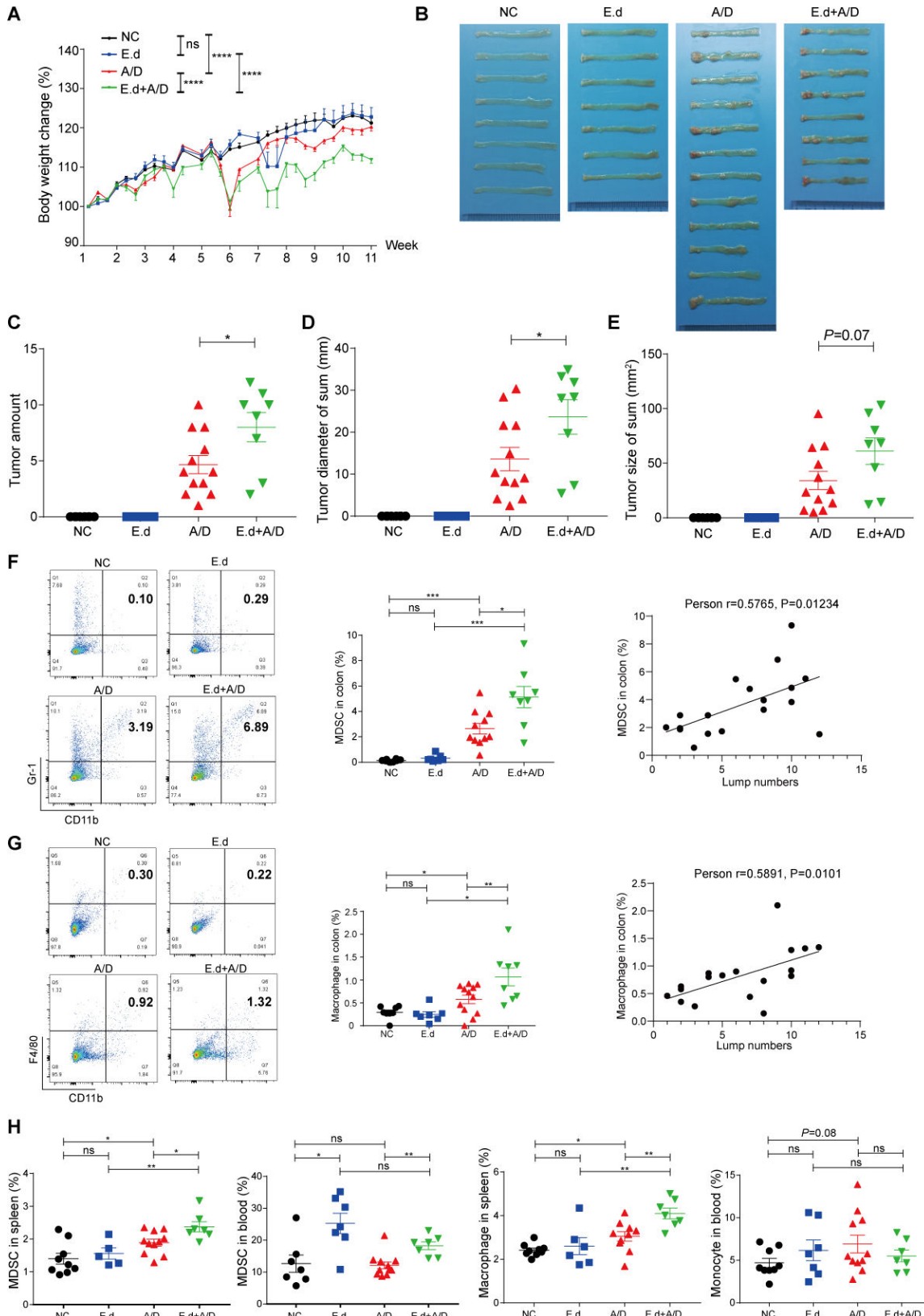

**FIG 3** E.d significantly exacerbated colonic inflammation and colon tumor formation in mice treated with AOM/DSS. We orally infected mice with E.d between AOM administration and DSS-induced tissue inflammation and compared those with mice undergoing the AOM/DSS regimen without external bacterial implantation. (A) Body weight change in mice after AOM injection (the first AOM injection was given on Day 1). (B) Images of lumps in the distal colons of mice (Continued on next page)

Fig 3 (Continued)

treated with AOM/DSS and 3 cycles of E.d. (**C–E**) Numbers (C), diameters (D), and sizes (E) of tumors at the end of the experiment when mice were sacrificed. (F, G) Relative abundances of MDSCs defined by CD11b$^+$ and Gr-1$^+$ (F) and macrophages defined by CD11b$^+$ and F4/80$^+$ (G) at the end of the experiment in the colons of mice (left panel: representative FACS scatter diagram; middle panel: FACS statistical results; right panel: correlation analysis between percentages of immune cells and tumor numbers). (H) Relative abundances of MDSCs and macrophages at the end of the experiment in the spleens and blood of mice. Data are expressed as the mean ± standard error of the mean; *$P < 0.05$; **$P < 0.01$; ***$P < 0.001$; ****$P < 0.0001$.

Master Mix (New England Biolabs, Hitchin SG4 0TY, UK), 0.2 µM forward and reverse primers, and 10 ng template DNA. Thermal cycling consisted of initial denaturation at 98$^\circ$C for 1 min, followed by 30 cycles of denaturation at 98$^\circ$C for 10 s, annealing at 50$^\circ$C for 30 s, and elongation at 72$^\circ$C for 60 s, and, finally, 72$^\circ$C for 5 min.

For 16S rRNA gene sequencing, a library was constructed using the NEB Next Ultra DNA Library Prep Kit for Illumina Library Construction from New England Biolabs. After qualification by Qubit and library detection, all amplicons were performed on an Illumina MiSeq platform by Novogene (Beijing, China) and Igenecode (Beijing, China). Fast length adjustment of short reads was used to merge paired-end reads. The QIIME software package (http://qiime.org/) and UPARSE pipeline (http://drive5.com/uparse/) were used to analyze the reads and pick operational taxonomic units (OTUs). Sequences with ≥97% similarity were assigned to the same OTUs. A representative sequence was picked for each OTU, and the Ribosomal Database Project classifier (rdp_classifier-2.2) (24) was used to assign taxonomic data to each representative sequence. Weighted UniFrac in QIIME was used to perform principal coordinate analysis (PCoA) (25).

## Flow cytometry

At the end of AOM/DSS induction, the mice were dissected, and the spleen, blood, and colorectal tissues were collected for the detection of related immune cells. For extracellular staining of immune markers, single-cell suspensions of immune cells were obtained from the spleen and blood tissue after lysis of red blood cells. Intestinal tissue is digested with collagenase (0.5 mg/mL, Sangon Biotech), collagenase D (1 mg/mL, Roche), dispase II (1 mg/mL, Roche), hyaluronidase (1 mg/mL, Sangon Biotech), and DNase I (0.2 mg/mL, Sangon Biotech) and ground to produce a single-cell suspension of immune cells. Tissue cells ($1 \times 10^6$/mL) were preincubated in a mixture of PBS, 1% bovine serum albumin, and different combinations of fluorochrome-coupled antibodies for 1 h. MDSCs were defined by CD11b$^+$ and Gr-1,$^+$ and macrophages were defined by CD11b$^+$ and F4/80$^+$. CD4$^+$ T cells defined by CD3$^+$ and CD4$^+$ and CD8$^+$ T cells defined by CD3$^+$ and CD8$^+$. Fluorescence data were collected with a FACS Canto II System (BD Biosciences) and analyzed with FlowJo software. The percentage of different cells is displayed based on different labels after the live cells are gated. The antibodies and reagents used for FACS were detailed in Table S1.

## Cell lines

The human colon cancer cell lines HCT116, HCT8, and DLD1 and the monocyte cell line THP-1 were obtained from the Cell Bank of the Type Culture Collection, Chinese Academy of Sciences (Shanghai, China). HCT116 cells were cultured in McCoy's 5A (Gibco) medium, and HCT8, DLD1, and THP-1 cells were cultured in RPMI-1640 (Gibco) medium supplemented with 10% fetal bovine serum (Gibco) and 1% penicillin/streptomycin antibiotics (Gibco). Cell lines in logarithmic growth phase were subcultured to maintain high cell viability.

## Western blotting

Cells were harvested by scraping into an SDS sample buffer containing a cocktail of protease and phosphatase inhibitors (Roche, Pleasanton, CA, USA). Western blotting was

conducted according to the standard procedure, and the related antibodies are shown in Table s2.

## Real-time PCR

### Stool sample

Mice stool samples were collected in a sterile microtube at the end of AOM/DSS induction, and total DNA was extracted from approximately 200 mg of mouse fecal material. Samples were stored at −80℃ until processed. All the extractions were performed by using OMEGA Stool DNA Kit (D4015-01) and following the manufacturer's protocol. DNA extracts were subjected to real-time quantitative PCR to reveal the relative abundance of R.f, F.s, and E.d. DNA concentrations were measured at 260 nm with a Nanodrop (3.0.1, ND- 2221000, Willington, USA), and purity was assessed by measuring the A260/A280. The input DNA was standardized and amplified for 40 cycles with SYBR Green Master Mix 224 (Invitrogen) and gene‐specific primers on a Roche Light Cycler 480 System (Roche, Basel, Switzerland). We used the 16S ribosomal RNA as the endogenous control. All primers used in the study are shown in Table S3.

### Colon tissue

Total RNA from mouse colon tissue was extracted using TRIzol reagent (Invitrogen, Grand Island, NY, USA), and then transcribed into cDNA using a Reverse Transcription System (Promega, Madison, WI, USA). The input cDNA was standardized and amplified for 40 cycles with SYBR Green Master Mix (Invitrogen) and gene‐specific primers on a Roche Light Cycler 480 System (Roche, Basel, Switzerland). We used the ACTB gene encoding β-actin as the endogenous control, and the samples were analyzed in triplicate. All primers used in this study are listed in Table s3.

## Statistical analysis

All statistical analyses were carried out using GraphPad Prism 8.0.2 (GraphPad Software, La Jolla, CA). Data are presented as the mean $\pm$ the standard error of the mean. Differences in parametric data between two groups were evaluated by Student's $t$-test (two-tailed). For correlation analysis between groups, Pearson's correlation coefficient test was used if the data fit a normal distribution; otherwise, Spearman's rank correlation coefficient test was applied. The significance of body weight changes was determined by one-way analysis of variance. Values of $P < 0.05$ were regarded as statistically significant, and significance is presented as *$P < 0.05$, **$P < 0.01$, ***$P < 0.001$, and ****$P < 0.0001$. Bonferroni adjustment with a statistical significance at $\alpha = 0.01$ was used for multiple testing corrections.

## RESULTS

### AOM/DSS failed to induce CRC in a small subset of mice despite severe inflammation

The standard AOM/DSS treatment regimen is highly effective in inducing CRC-like colonic adenocarcinoma in mice within 2–3 months. A total of 44 mice were initially treated with AOM and three cycles of 1.5% DSS to induce CRC (Fig. 1A) (26). Among them, 23 mice survived the full treatment regimen, in which 20 developed noticeable tumors/lumps (A/D-T group), and three did not (A/D-NT group) by the uniform end-point (Fig. 1B and C). Therefore, a highly effective CRC induction methodology is demonstrated by a close to 50% lethality rate (21/44 mice had to be euthanized prior to the protocol endpoint), and numerous uniform CRC resembling histological features within colorectal tissues in every mouse developed tumors, including the full spectrum of epithelial cell polarity loss, aberrant crypt foci, inflammatory edema, and submucosal infiltration of inflammatory immune cells (Fig. 1D). Importantly, while the absence of apparent tumors in all three A/D-NT colons was confirmed with the absence of aberrant crypt foci

formation under close microscopic and histological examinations, the absence of tumors in these three animals did not reflect a failed attempt to induce a chronic inflammatory gut since significant mucosal ulcerations and immune infiltration were uniformly observed (Fig. 1D) throughout the entire colon in every histology sample of DSS-treated animals, regardless of whether tumors developed. In stark contrast with the healthy gut histology of control mice, we conclude that the AOM/DSS treatment regimen uniformly induced severe colonic inflammation, while a small minority of mice were spared from tumors (Fig. 1D). Although we cannot exclude the possibility of AOM-induced genotoxic effects being nonspecific in nature and, therefore, not hitting enough critical CRC-related tumor driver genes in every treated animal, a more attractive scenario might be that distinctive gut microbial structural differences exist among different animals. A rare, peculiar combination of gut microbial flora might be responsible for the emergence of extreme tumor resistance despite combined harsh damages caused by genotoxic insults, followed by chronic epithelium inflammation. To verify whether this is indeed the case, feces were collected from three mice without AOM/DSS treatment (NC group), three mice with severe tumor formation (A/D-T group), and three mice with no detected tumor formation (A/D-NT group) for detailed gut microbiota profiling.

## Distinctive gut microbiota structures discovered in CRC-free mice

To investigate the role of the gut microbiota in CRC development and identify differences in the microbiota among the three groups (NC, A/D-T, and A/D-NT) of mice, sequence analysis via 16S rDNA high-throughput sequencing was performed (27). Despite the lack of species-level distinction power in many cases, 16S rDNA metagenomics analysis is highly effective in obtaining the overall gut microbiome structural features. While observed inter-individual microbiota variation and our limited sample size complicate the analysis, the Chao1 index and observed species number obtained from the alpha diversity analysis in the A/D-NT group were significantly higher than those in the A/D-T group ($P = 0.0269$, and $P = 0.0348$, respectively) (Fig. 4A), approaching those observed in the NC group ($P = 0.7056$ and $P = 0.5964$, respectively). With weighted UniFrac PCoA of OTU abundance, there was clear separation between samples from the A/D-T group and the other two groups, while those from the A/D-NT and NC groups are largely overlapping on the plot (Fig. 4B). These results collectively indicated an overall similarity in gut microbial diversity between the NC and AD-NT groups, while those from A/D-T demonstrated overall decreased gut microbial diversity and shifted microbial communal structures. Such a result is also consistent with prior findings that certain commensal bacteria restrict epithelial over-proliferation (16). Therefore, the loss of particular commensal bacteria during chronic colonic inflammation removes corresponding check and balance mechanisms, and, subsequently, tumors develop. To pinpoint which residual commensal bacteria species might be responsible for rare tumor-free status in AOM/DSS-treated animals, we carried out LEfSe analysis to identify differential bacteria abundance on the OTU level. Such analysis indicates that OTUs closely related to the S24-7 family (e.g., *Prevotellaceae*, *Rikenellaceae*, *Methanomicrobiaceae*, *Tenericutes*, *Allobaculum*, and *Ruminococcus*) were significantly higher in the A/D-NT group compared with their tumor-bearing counterparts. The critical CRC protective commensal bacteria might be one or more particular species from those OTUs. On the contrary, the abundance of OTUs related to *Proteobacteria*, *Erysipelotrichi*, *Coprobacillaceae*, *Lactobacillaceae*, and *Enterobacteriales* was much higher in the A/D-T group (Fig. 4C D). Considering potential utilization for CRC prognosis and prevention, differences among the three groups were further analyzed at the species level (Fig. 4E F). We found a significant decrease in *R. flavefaciens* ($P = 0.0003$) and *F. succinogenes* ($P = 0.0012$) in the feces of A/D-T mice compared to A/D-NT mice, suggesting these two as strong CRC protective bacterium candidates. Although R.f and F.s abundance seems to be decreased in A/D-NT animals compared with the NC group, the degree of drop is much less severe compared with the extent of R.f and F.s disappearance in the A/D-T group. In addition, while E.d was barely detectable in both NC and A/D-NT mice, it is highly abundant in

all tumor-bearing animals, suggesting it as a potential prognostic microbial marker for CRC occurrence (Fig. 4G). Furthermore, consistent results were validated on more animals with RT-PCR (Fig. S1).

## *E. dolichum* promotes CRC development course

E.d was barely detected in fecal samples from the NC group and mice with no tumor formation (A/D-NT group), but it was abundant in fecal samples from mice with tumors (Fig. 4G). Such observation indicates that E.d is specifically amplified in the inflammatory gut prone to tumor development. Besides being a potential biomarker for CRC risk, we also asked the question whether E.d proliferation directly facilitates tumor progression. To this end, we orally infected mice with E.d between AOM administration and DSS-induced tissue inflammation and compared those with mice undergoing the AOM/DSS regimen without external bacterial implantation. E.d administration alone had no effects on body weight compared to control mice, suggesting E.d implantation did not cause significant physiological changes to mice by itself (Fig. 3A). However, mice that underwent E.d implantation exhibited more drastic rapid weight loss after DSS feeding in the CRC modeling group than mice that underwent AOM/DSS treatment alone (Fig. 3A), suggesting E.d was specifically detrimental to animal health during colonic inflammation. Compared to AOM/DSS treatment alone, additive E.d treatment also exacerbated tumor formation as determined by bright field imaging of colons (Fig. 3B), numbers of tumors per animal, and average tumor size (Fig. 3C through E). Additive E.d treatment significantly increased the numbers of MDSCs and macrophages in the colon compared to AOM/DSS treatment alone, and there were significant positive correlations (Fig. 3F and G) between either MDSC and macrophage quantities with the number of tumors in each animal. Similar trends were observed in the numbers of MDSCs in the spleen and blood, as well as the number of macrophages in the spleen (Fig. 3H and Fig. S2A through D), consistent with severe colon-centered inflammation spreading over to the whole organism. These increases in MDSCs and macrophages were not associated with corresponding changes in proportions of CD4$^+$ or CD8$^+$ T cells either in the colon or in circulation (Fig. S2E and F). In addition, we found that E.d administration increased the expression of the pro-inflammatory cytokines, including IL-1β, IL-2, IL-6, TNF-α, and IFN-γ in the animals, regardless of AOM/DSS treatment, though some of which failed to reach statistical significance (Fig. 5A). Overall, these immunological features are consistent with a locally exacerbated gut inflammatory environment during repeated tissue damages when significant amounts of *E. dolichum* are present, with minimal T cell infiltration into the damaged tissue. While the boosted immunity reactivity is restricted to the innate immunity cells, there seems to be a good indication that animals with the most severe colon inflammation status are more likely to develop the greatest number of colonic adenocarcinomas, which is consistent with the recent clinic finding that higher inflammation after diagnosis was significantly associated with worse survival outcomes among patients with stage III colon cancer (28).

On top of validating that E.d is indeed a pivotal microbe closely associated with CRC development, we directly investigated whether E.d can potently activate innate immunity in the gut. Both gut epithelial cells and innate immune cells can sense various pathogen-associated molecular patterns (PAMPs) and trigger downstream signaling transduction pathways to elicit inflammatory responses. Heterodimers of p50 and p65 (RelA) mediate the most abundant form of canonical NF-κB pathway, which depends on the transient p65 phosphorylation in response to PAMP sensing, subsequent trafficking into the nucleus, and transcription activation on a variety of immune effector genes. We found adding E.d culture onto cultured human colon cancer cell line HCT116 triggered a robust p65 phosphorylation, peaking about 0.5 h after inoculation (Fig. 5B). This effect is dosage-dependent and can be triggered as efficiently by its culture supernatant alone, suggesting E.d can secrete potent PAMP molecules in gut lumen to activate NF-κB dependent inflammatory response (Fig. 4C and D). We also compared the effect of E.d culture supernatant in activating human colorectal adenocarcinoma epithelial cells or

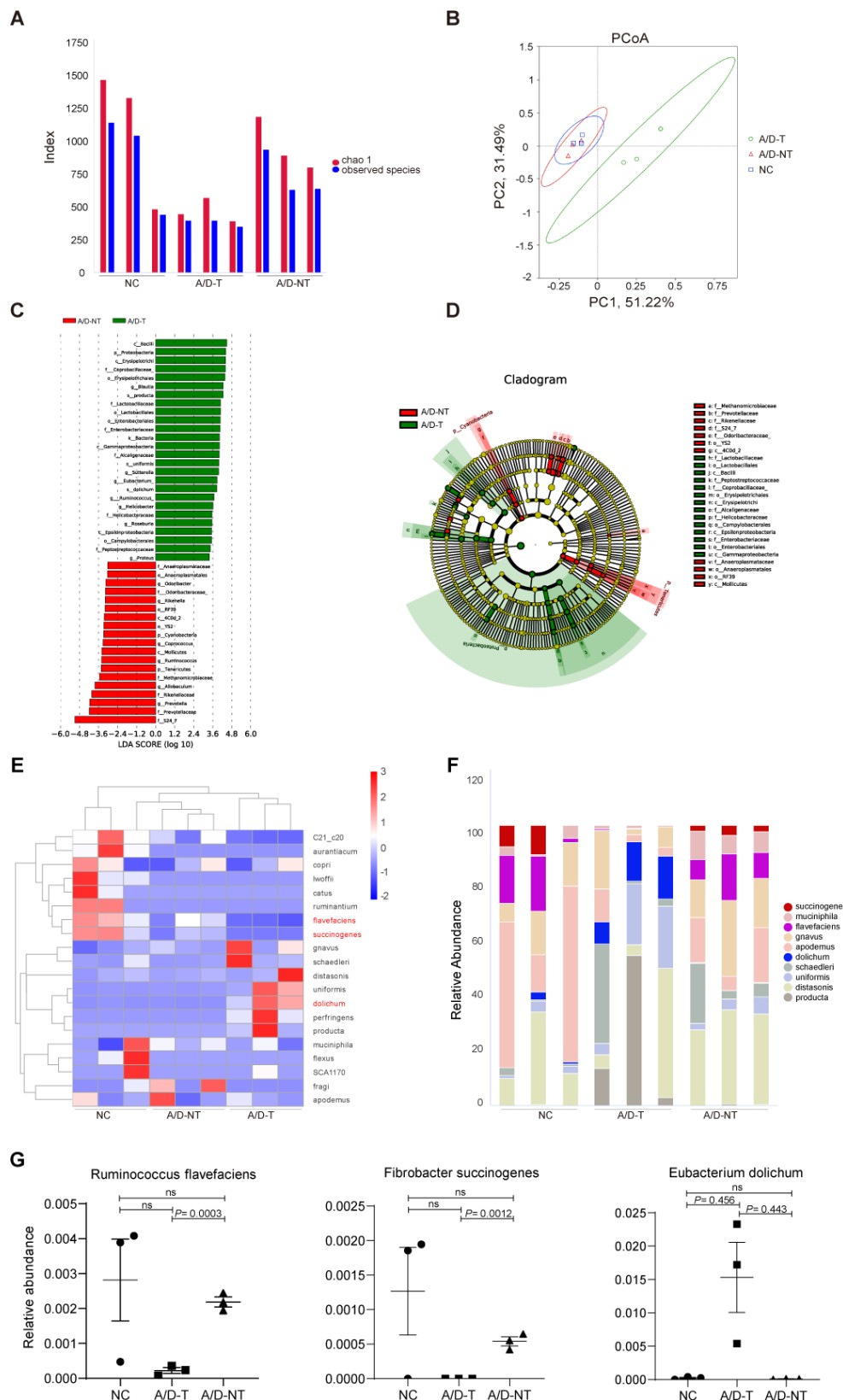

FIG 4  Relative abundances of gut microbes in mice. (A) Analysis of the alpha diversity correlation index of Chao1 and observed species. (B) Clustering of microbial communities using PCoA. Percentage values at the axes indicate the contribution of principal components to the total variance in the data set. (C, D) LEfSe analysis of differences in abundance between mice (Continued on next page)

**Fig 4 (Continued)**

with and without tumors. (C) Histogram of linear discriminant analysis (LDA) value distribution. (D) Evolutionary branching diagram. (E, F) Relative abundance at the species level in fecal samples from mice. (E) Histogram of relative abundance of species (three differential species are shown in red). (F) Clustering heat map of species abundance. (G) Relative abundance of three differential species in fecal samples from mice. Data are expressed as the mean ± standard error of the mean. $n = 3$.

human monocytes. Monocyte cell line THP-1 can be more potently activated by minimal amounts of E.d culture supernatant compared with colon cancer cell lines DLD1 and HCT8, suggesting that residual innate immune cells in the gut are particularly sensitive to the presence of E.d (Fig. 5E).

Since aberrant activation of NF-κB has been observed in a wide range of human cancers and is thought to promote tumorigenesis and metastasis, we speculate E.d's chronic proliferation in the inflamed gut will promote tumorigenesis through aberrant activation of NF-κB. In many chronic inflammatory diseases, activated p65 protein failed to undergo proteolytic degradation to attenuate the inflammatory signaling back to baseline level. Therefore, accumulation of p65 at a high abundance level leads to constitutive NF-κB activity and contributes to the progression of cancer (29–31). Indeed, we observed a generally elevated p65 protein level from colon extract taken from mice undergoing AOM/DSS treatment. Implantation of E.d to mice without eliciting DSS-mediated inflammation also caused an increase in steady-state p65 level ($P = 0.0655$) (Fig. 5F), consistent with our finding that it is a potent agent for NF-κB activation. Although there was a significant difference between p65 steady-state levels with or without E.d inoculation, the extent of p65 level increase caused by E.d inoculation alone is rather modest. However, p65 levels from E.d pre-implanted, AOM/DSS-treated gut were consistently drastically elevated, especially when compared with AOM/DSS-treated mice without prior E.d inoculation ($P = 0.0006$) (Fig. 5F). Such additive effect on chronic p65 activation provided an emerging scenario that E.d promoted CRC through being a potent, chronic trigger for aberrant NF-κB activity potentially for both the innate immunity cells and the gut epithelial cells themselves. Such prolonged intertwined interactions between NF-κB-activated epithelial and inflamed innate immune cells, therefore, favor CRC development.

## R.f or f.s are protective commensal bacteria preventing tumorigenesis in the colon

While identification of the causal role of E.d in CRC tumorigenesis offers an attractive cancer predictive marker, it is challenging to remove a particular bacterium from the human gut to prevent disease progression. Therefore, we switched our attention to R.f and F.s, which were found to be persistent in abundance in CRC-free AOM/DSS-treated mice but largely lost in animals that developed cancer. Our hypothesis is that one or both of those act as critical protective commensal bacteria, whereas cancer progression can be prevented or delayed if they are present in the gut with high abundance. Therefore, we took a similar approach to preimplant R.f or F.s in the animals to be treated with AOM/DSS and evaluated whether those bacteria can demonstrate protective effects in terms of developing colon cancer. In contrast to severe body weight loss of E.d-implanted animals during the DSS treatment period, intragastric administration of R.f or F.s affected neither the baseline body weight with untreated mice nor the body weight fluctuation during DSS treatment (Fig. 2A F). Therefore, R.f and F.s appeared to be true commensal bacteria without inducing gut inflammation as E.d did (Fig. 3A). Additive R.f. implantation had a slight tendency to influence the reduction of tumor formation, although the effect was not statistically significant with limited numbers of animals we experimented (Fig. 5B through E). Nevertheless, we found that pre-implanting R.f significantly decreased the numbers of blood MDSCs ($P < 0.0001$) and macrophages ($P < 0.0001$), even in the animals undergoing AOM/DSS treatment ($P = 0.0110$ for MDSC and $P = 0.0042$ for macrophage) (Fig. S3A and B). This reduction of inflammatory signature might not be dramatic enough to counterbalance the severe DSS-triggered

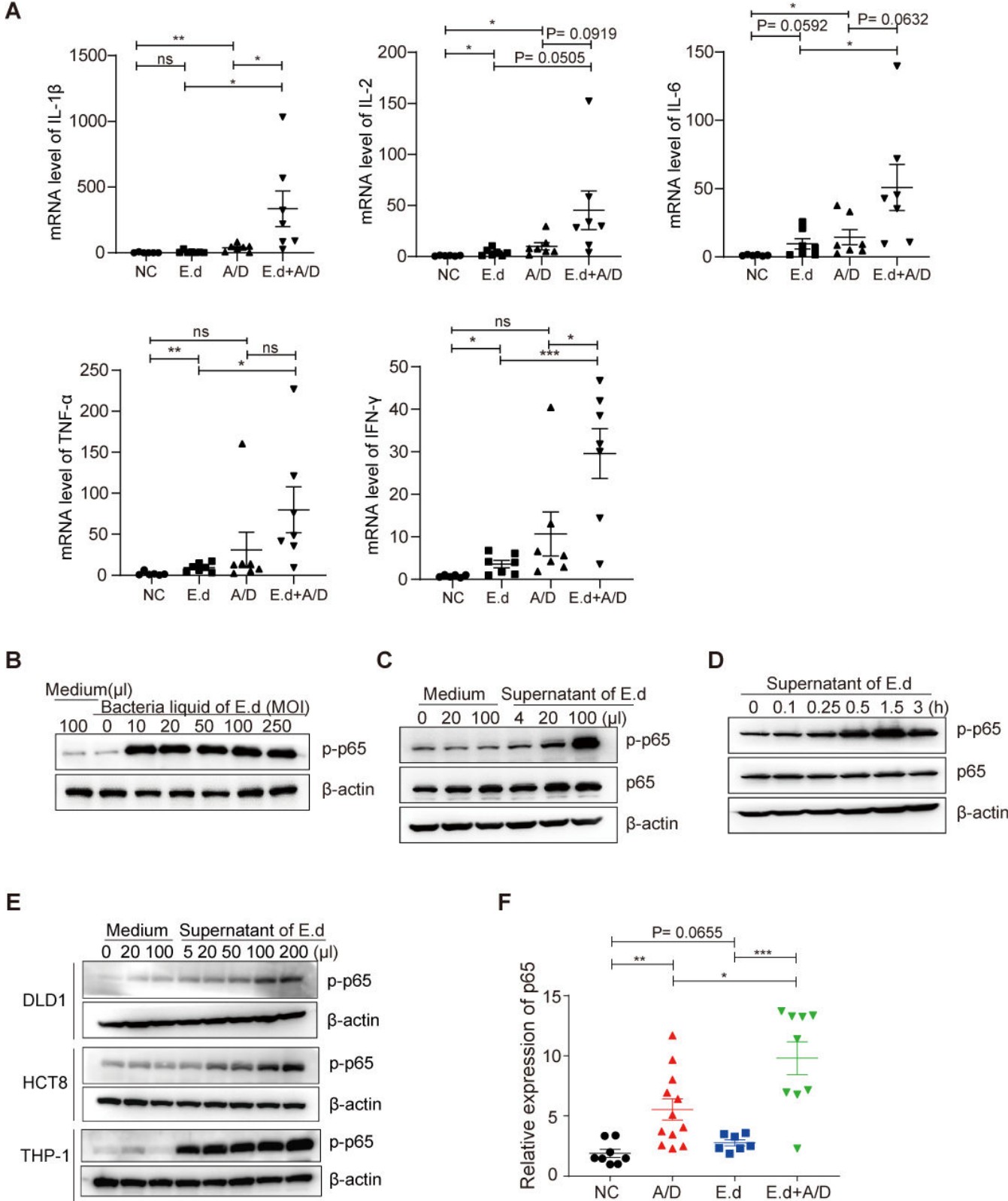

**FIG 5** E.d aggravated intestinal inflammation and activated the NF-κB pathway. (A) After the mice were sacrificed, the intestinal tissues were collected and homogenized, and quantitative PCR was performed to detect the relative expression levels of several pro-inflammatory cytokines (IL-1β, IL-2, IL-6, TNF-α, and IFN-γ). (B) P-p65 and p65 protein expression in HCT116 cells treated with different concentrations of control medium or E.d supernatants for 0.5 h. (C) P-p65 and

Fig 5 (Continued)

p65 protein expression in HCT116 cells treated with 200 µL of E.d supernatants at different timepoints. (D) P-p65 and p65 protein expression in HCT116 cells treated with different concentrations of bacterial suspension and bacteria liquid of E.d for 0.5 h. (E) P-p65 and p65 protein expression in DLD1, HCT8, and THP-1 cells treated with different concentrations of control medium or E.d supernatants for 0.5 h. (F) p65 protein expression in the colon tissue of AOM/DSS-induced mice treated with E.d (left panel: western blotting; right panel: statistical results). Data are expressed as the mean ± standard error of the mean. *$P < 0.05$; **$P < 0.01$; ***$P < 0.001$.

inflammation in the colon; therefore, we did not observe a statistically meaningful reduction of MDSC and macrophage counts in the colon as seen in the blood (Fig. S3C and D). However, there are general, significant positive correlations (Fig. S3A and B) that still hold true between either inflammatory cell type quantities with the number of tumors in each animal. In addition, we found R.f administration reduced the expression of the pro-inflammatory cytokines, including IL-1β ($P = 0.0184$), IL-6 ($P = 0.0566$), TNF-α ($P = 0.0167$), and IFN-γ ($P = 0.0850$) in the animals undergoing AOM/DSS treatment (Fig. S5A).

Given such close correlation between inflammation severity and CRC tumor burden as our mouse study verified, we argue R.f is a protective commensal bacterium in the gut due to its anti-inflammatory mode of function, even though supplementing R.f alone might not significantly switch divergent disease outcomes under our extremely severe inflammation-triggered mouse CRC model.

Devising protective commensal bacteria-based probiotics to influence colon cancer disease outcome is a critical long-term goal to achieve. We envision that with our acute AOM/DSS-induced CRC model, the severity of inflammatory insults toward the colonic epithelium would greatly surpass those during typical human disease progression. Therefore, a significant reduction of cancer burden with a single bacteria inoculation within the context of such artificial mouse cancer model can be transferred to a strongly validated rationale for a similar probiotics-based colon cancer prevention approach in humans. After the initial trial with *Ruminococcaceae* implantation without reaching statistically divergent disease outcomes, we turned our attention to *F. succinogenes*, which is a species belonging to the *Fibrobacteraceae* family, mainly found in ruminant animals for cellulose degradation. Due to the difference in dietary structure between rodents and humans, F.s is rarely discovered in the human digestive tract (32). Nevertheless, since F.s was abundant in the gut microbiome of CRC-free AOM/DSS-treated mice, we evaluated whether implanting F.s can prevent or delay colon cancer progression in our CRC mouse model. To our great satisfaction, we observed a significantly reduced number of tumors ($P = 0.0111$) formed in AOM/DSS-treated mice if F.s is pre-implanted (Fig. 2G). The tumors formed are also, in general, much smaller in size ($P = 0.0102$) compared with those in the control AOM/DSS-treated group (Fig. 5H through J). Therefore, retaining F.s in the colon was significantly functional in blocking colon cancer development in mice. We noticed, in contrast to R.f, that F.s treatment had no effects on the proportions of MDSCs or macrophages in the colon, spleen, or blood compared to treatment with AOM/DSS alone (Fig. S4A and B). The pro-inflammatory cytokine (IL-1β, IL-2, IL-6, TNF-α, IFN-γ) in intestinal tissues did not change significantly with the administration of F.s (Fig. S5B). Therefore, F.s is not an anti-inflammatory commensal bacterium as R.f. However, we notice a significant increase ($P = 0.0196$) of colorectal CD4$^+$ T cells and a non-significant increase ($P = 0.0534$) of colorectal CD8$^+$ T cells in F.s-implanted mice under AOM/DSS treatment (Fig. S4C and D). Further detailed studies would be needed to elucidate how F.s modulates gut immune microenvironment and whether this creates a favorable environment for T cells and possibly facilitate innate immune cells to enter the inflammatory site surveying metastatic cancer cells generated by prior genotoxic insults. This intriguing linkage between gut microbes, colorectal T lymphocytes, and cancer surveillance makes this model a promising arena for mechanistic studies aiming for critical translational cancer prevention.

This clear cancer preventative effect by F.s within a severe inflammation-triggered colon cancer model strongly validates that a probiotics-based colon cancer prevention

strategy can be clinically effective soon. Although transferring protective commensal bacteria for human digestive disease intervention awaits much further mechanistic and clinical studies, our current investigation using a well-established, severe inflammation-induced murine CRC model collectively demonstrated that maneuvering a small number of critical gut flora can greatly influence the disease outcome, including whether within a highly inflammatory gut, cancer will develop or not.

## DISCUSSION

The human intestine harbors a vast number of bacterial cells, ranging from $10^{13}$ to $10^{14}$, along with over 500 different types of microorganisms (33). As the result in previous studies (16), germ-free mice subjected to the AOM/DSS model often exhibit significantly reduced or even absence of tumor formation. This is likely due to the critical role of the gut microbiota in modulating inflammation and thereafter tumorigenesis. The gut microbiome is a complex and diverse microcosmos, making it unlikely that a single bacterial population is solely responsible for driving or preventing tumorigenesis. Although there has been no specific, defined microbial community composition associated with the development of CRC, certain associative patterns can serve as risk markers, potentially aiding early diagnosis. On the contrary, the causal link between the rise or fall of a particular bacteria species in the gut and CRC development has been extremely difficult to establish. Although chronic inflammation has been known for a long time as a critical prerequisite for CRC, only a small fraction of IBD patients develop CRC, suggesting a missing regulatory link between IBD and CRC can be greatly influenced by additional factors, including individual microbiota. - In our study, a small subset of mice did not develop CRC when exposed to AOM/DSS, an extremely potent combination with mutagenesis and inflammation inducing CRC. We employed high-throughput 16S rRNA sequencing to compare the microbial community compositions in the feces of these mice. Our analysis suggested that distinctive changes in the gut microbiome are closely associated with the end result of whether CRC is developed or not with this isogenic animal model. R.f and F.s were markedly abundant in mice without CRC formation, whereas E.d was only detected in mice with CRC formation. These results are consistent with previous reports (12, 34) that observed higher levels of *Eubacterium* in CRC patients compared to the healthy population (12). While such observations are consistent with the hypothesis that specific bacterial species might have particularly prominent effects on human CRC tumorigenesis, a large patient cohort will be necessary in the future, especially by recruiting in parallel both CRC-free IBD patients and those who have developed CRC from chronic gut inflammation.

With our mouse model, however, it is possible to establish a critical putative causal relationship between a residual microorganism and the transition from chronic inflammation to CRC. Preimplantation of all three bacteria does not significantly cause gut inflammation by itself as triggered by the DSS treatment regime. However, we observed clear, predictive CRC development rate shifts by introducing different microorganisms in the gut before chemically induced carcinogenesis. Therefore, the key microbiota divergence we observed comparing CRC-free and CRC mice under identical treatment regime can be attributed at least in part retrospectively to functional contributions from individual bacteria species. Although we cannot rule out the possibility of strong synergetic interactions from multiple residual bacteria species during chronic inflammation, our single species manipulations, without knowing corresponding full-scale impacts to the remaining microbiota structure in the colon, do suggest distinctive contributions of three different microorganisms. E.d is known to produce lactic acid, acetic acid, propionic acid, and butyric acid, which can affect the gut microbiota and host metabolism (35, 36). Dietary fiber intake has been shown to influence the composition of the intestinal flora, thereby affecting the risk of cancer. In a study investigating relationships between fiber intake and gut microbial community composition and taxa abundance, higher fiber intake was weakly correlated with a higher abundance of *Faecalibacterium prausnitzii* and a lower abundance of E.d (37). E.d

treatment exacerbated tumor formation by significantly increasing both tumor numbers and tumor size per animal. Such E.d-associated detrimental effects can even be observed by severe, persistent animal body weight loss upon induction of colon inflammation, suggesting the presence of a significant amount of E.d in an inflamed gut is highly disruptive and greatly exacerbates CRC development. Similar treatment with R.f or F.s demonstrated preventative effects on CRC development with different extents, with F.s implantation showing a significant reduction of both tumor number and tumor size on treated animals. R.f, F.s, and *Ruminococcus albus* are commonly known as the three main microorganisms involved in fiber decomposition in the rumen (38, 39). F.s is as effective as *Ruminococcus* in degrading plant cell walls (40), although they utilize different metabolic means. The cellulase degradation mediated by these cellulase-producing bacteria may have an additive, protective effect against CRC development (41). While this clear protective role of F.s in CRC prevention cannot be readily transferred to humans, it clearly demonstrated that CRC development can be blunted effectively within a chronic inflammatory gut with the implantation of even a single beneficial bacteria species. Therefore, a probiotic-centered preventative strategy for CRC management among IBD patients is likely to be fruitful once we can establish solid causal cancer prevention function, as well as deep mechanistic understanding of specific commensal bacteria in IBD patients parallel to what we have accomplished with the current inflammation-induced CRC mouse model.

Our findings also suggested that distinct microorganisms could shape the immune microenvironment in the inflamed gut differently, which could be a critical factor determining the endogenous tumor surveillance activity of the immune system. Among our studies, we observed a striking positive correlation (Fig. S3A and B) across all experimental groups between either MDSC and macrophage quantities with the number of tumors in each animal. This is consistent with previous publications on MDSCs, taking a critical role in the development of AOM/DSS-induced murine CRC (26) potentially by suppressing cytotoxic CD8[+] T cell infiltration (42). E.d drastically promoted the accumulation of both MDSCs and macrophages in DSS-treated gut via activating the NF-κB signaling pathway, part of which leaves the digestive tract and enters the circulation. Pre-implanting R.f had an opposite role in suppressing MDSCs and macrophages. This effect is rather modest to counterbalance the severe DSS-triggered inflammation in the colon, but R.f significantly decreased the numbers of blood MDSCs and macrophages, which reflects a less severe overall inflammatory signature. Although F.s demonstrates a similar CRC blocking effectiveness as R.f, our analysis implies that their mode of action could be divergent. F.s is not an anti-inflammatory commensal bacterium like R.f since it has no effects on the proportions of MDSCs or macrophages in the colon, spleen, or blood compared to treatment with AOM/DSS alone (Fig. S4A and B). However, we found a significant increase in colorectal CD4[+] T cells and an equivalent increase in CD8[+] T cells in F.s -pre-implanted animals, suggesting F.s can facilitate T cell tissue infiltration in the presence of suppressive MDSCs. Whether some of these T cells were responsible for active removal of cancer cells once generated would be an important area of future investigations. Regardless of the detailed mechanism of cancer-immunity interplay in the inflamed gut, our results clearly showed some microorganisms can break the MDSC-imposed suppressive immune environment to promote effective immune surveillance, although their exact mode of influence on the gut immune environment awaits further investigations. Nevertheless, this area of research is critically important for combating CRC, a cancer type that is overwhelmingly deemed "immune cold" and responds poorly to a variety of current immune checkpoint blockade therapies (43). Since F.s is rarely found in the human digestive tract, further work is needed to identify human commensal bacteria that can reshape the inflamed gut immune environment in a similar manner.

## ACKNOWLEDGMENTS

This work was supported by grants to W.H. from the National Natural Science Foundation of China (82121004, 81790254, 91629301 and 81872354) and the Major State Basic Research Development Program of China (2013CB910802).

W.H. conceived and supervised the project; W.H., F.G., and D.Z. wrote the manuscript; D.Z., Y.S., and P.D. performed most of the experiments and analyzed the data with assistance from X.W., Ling Li, Luying Li, X.L., T.L., J.C., W.Z., and Q.W.; and Q.J., F.G., and W.H. contributed ideas.

The authors declare that the research was conducted in the absence of any commercial or financial relationships that could be construed as a potential conflict of interest.

## AUTHOR AFFILIATIONS

[1]Institutes of Biomedical Sciences, Shanghai Medical College, Fudan University, Shanghai, China

[2]Fudan University Shanghai Cancer Center, Shanghai Medical College, Fudan University, Shanghai, China

[3]Department of Head and Neck Surgery, Fudan University Shanghai Cancer Center, Shanghai, Shanghai, China

[4]Department of Oncology, Shanghai Medical College, Fudan University, Zhengzhou, China

[5]Department of Neuroimmunology, Zhengzhou University, Zhengzhou, Henan, China

[6]Henan Engineering Technology Research Center for Accurate Diagnosis Neuroimmunity, Zhengzhou University, Zhengzhou, China

[7]Henan Institute of Medical and Pharmaceutical Sciences, Zhengzhou University, Zhengzhou, China

[8]Key Laboratory of Breast Cancer in Shanghai, Fudan University Shanghai Cancer Center, Fudan University, Shanghai, China

## AUTHOR ORCIDs

Danlei Zhou  http://orcid.org/0000-0003-0616-8712
Weiguo Hu  http://orcid.org/0000-0002-7397-6800

## FUNDING

| Funder | Grant(s) | Author(s) |
| --- | --- | --- |
| National Natural Science Foundation of China | 82121004 | Weiguo Hu |
| National Natural Science Foundation of China | 81790254 | Weiguo Hu |
| National Natural Science Foundation of China | 91629301 | Weiguo Hu |
| National Natural Science Foundation of China | 81872354 | Weiguo Hu |
| Major State Basic Research Development Program of China | 2013CB910802 | Weiguo Hu |

## AUTHOR CONTRIBUTIONS

Yujing Sun, Data curation, Investigation, Resources, Writing – review and editing | Peipei Ding, Data curation, Investigation, Methodology, Writing – review and editing | Xiaochao Wang, Data curation, Investigation, Writing – review and editing | Ling Li, Data curation, Investigation, Methodology, Writing – review and editing | Luying Li, Data curation, Investigation, Writing – review and editing | Xinyue Lv, Data curation, Investigation, Writing – review and editing | Tian Liao, Data curation, Investigation, Writing – review and editing | Jianfeng Chen, Data curation, Investigation, Writing – review and editing | Wei Zhang, Data curation, Investigation, Writing – review and editing | Qi Wang, Data curation, Investigation, Writing – review and editing | Qing-Hai Ji, Data curation, Supervision, Writing – review and editing | Feng Gao, Methodology, Supervision, Writing

– review and editing | Weiguo Hu, Conceptualization, Data curation, Funding acquisition, Project administration, Resources, Supervision, Writing – review and editing.

## DATA AVAILABILITY

The microbiome sequencing data sets for this study can be found in the Sequence Read Archive (SRA) database at NCBI under BioProject ID PRJNA799170.

## ETHICS APPROVAL

The Animal Ethics Committee of Shanghai Medical School approved all animal experiments.

## ADDITIONAL FILES

The following material is available online.

### Supplemental Material

**Figure S1 (Spectrum02792-24-S0001.jpg).** Relative abundance of (A) *Ruminococcus flavefaciens*, (B) *Fibrobacter succinogenes*, and (C) *Eubacterium dolichum* in fecal samples from mice.
**Supplemental figure legends and tables (Spectrum02792-24-S0002.docx).** Supplemental figure legends and Tables S1 to S3.
**Figure S2 (Spectrum02792-24-S0003.jpg).** Relative abundances of immune cells in mice treated with AOM/DSS and 3 cycles of E.d.
**Figure S3 (Spectrum02792-24-S0004.jpg).** Relative abundances of immune cells in mice treated with AOM/DSS and 4 cycles of R.f.
**Figure S4 (Spectrum02792-24-S0005.jpg).** Relative abundances of immune cells in mice treated with AOM/DSS and 3 cycles of F.s.
**Figure S5 (Spectrum02792-24-S0006.png).** Relative abundances of pro-inflammatory markers in a mouse colon after AOM/DSS induction and R.f/F.s administration.

### Open Peer Review

**PEER REVIEW HISTORY (review-history.pdf).** An accounting of the reviewer comments and feedback.

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
