## [Reviewer comments · Microbiology Spectrum]

Microbiology Spectrum

Gut Commensal Bacteria Influence Colorectal Cancer Development by Modulating Immune Response in AOM/DSS-treated Mice

Danlei ZHOU, Yujing Sun, Peipei Ding, Xiaochao Wang, Ling Li, Luying Li, Xinyue Lv, Tian Liao, Jianfeng Chen, Wei Zhang, Qi Wang, Qinghai Ji, Feng Gao, and Weiguo Hu

Corresponding Author(s): Danlei ZHOU, Fudan University

Review Timeline:

Submission Date:	November 11, 2024
Editorial Decision:	February 5, 2025
Revision Received:	March 23, 2025
Accepted:	March 30, 2025

Editor: Jan Claesen

Reviewer(s): Disclosure of reviewer identity is with reference to reviewer comments included in decision letter(s). The following individuals involved in review of your submission have agreed to reveal their identity: Huoxiang Zhou (Reviewer #2)

Transaction Report:

DOI: <https://doi.org/10.1128/spectrum.02792-24>

Re: Spectrum02792-24 (Gut Commensal Bacteria Influence Colorectal Cancer Development by Modulating Immune Response in AOM/DSS-treated Mice)

Dear Ms. danlei zhou:

Thank you for the privilege of reviewing your work. Below you will find my comments, instructions from the Spectrum editorial office, and the reviewer comments.

Thank you for submitting your research to Spectrum. Your work has now been evaluated by three independent Reviewers, who are generally enthusiastic (as am I). The Reviewers pointed out some comments and suggestions to help improve the manuscript and I would be happy to consider a revised version which addresses these comments in a point-by-point manner. In addition, please make sure the associated data (for sequencing) is deposited in a public repository and referred to in the data availability statement (please see the ASM guidelines for more information). In addition, make sure that limitations pointed out by the Reviewers are appropriately discussed and it might be beneficial to compare your findings and conclusions to studies in the literature dealing with (human) colorectal cancer microbiome.

Revision Guidelines

Sincerely,
Jan Claesen
Editor
Microbiology Spectrum

Reviewer #1 (Comments for the Author):

This study explores the role of gut microbiota in colorectal cancer (CRC) development using an AOM/DSS-induced mouse model. The authors identified *Eubacterium dolichum* (E.d.) as a CRC-promoting bacterium and *Ruminococcus flavefaciens* (R.f.) and *Fibrobacter succinogenes* (F.s.) as potential protective bacteria. While the findings are interesting and contribute to the growing understanding of microbiota in CRC, the study has several limitations that need to be addressed to strengthen its conclusions and translational relevance.

1. The study relies on a small number of mice ($n = 3$ per group for microbiota sequencing), which raises concerns about statistical robustness.
2. Do the microbiota identified from the mouse study also apply to human colorectal cancer?
3. Did the authors use a germ-free mouse model to assess the impact of specific bacteria on tumor development and progression?
4. The study suggests NF- κ B activation as a key pathway, but mechanistic validation is minimal. Transcriptomic or cytokine profiling in colonic tissues could provide stronger mechanistic support.

Reviewer #2 (Public repository details (Required)):

Please provide the ID of 16S rRNA gene sequencing data in a public repository.

Reviewer #2 (Comments for the Author):

The manuscript investigated the gut microbiota in AOM/DSS treated mice with and without CRC formation. The results revealed that *Ruminococcus flavefaciens* and *Fibrobacter succinogenes* were significantly enriched in CRC-free mice, whereas the presence of *Eubacterium dolichum* was dramatically reduced. Bacterial intragastric administration experiments demonstrating protective roles of *Ruminococcus flavefaciens* and *Fibrobacter succinogenes*, and detrimental role of *Eubacterium dolichum* in CRC initiation. This paper provides information about effects of gut commensal bacteria on CRC development and it is helpful for the readers on this area. However, there are limited novelty and less mechanism in this study. In addition, some questions in this manuscript should be addressed.

Major:

1. 16S analysis is not informative enough as to understanding the mechanistic details. None of the results from correlation analysis is confirmed by experiments.
2. Antibiotics like metronidazole are difficult to dissolve in water, and How is the antibiotic cocktail prepared? Antibiotics have a very strong smell and caused mice to dislike drinking water. Why not use intragastric treatment?
3. While the manuscript presents many statistical comparisons (e.g., between NC, A/D-T, and A/D-NT groups), it would be helpful to provide more details on the statistical methods used, especially for complex analyses like PCoA, OPLS-DA, and Spearman's correlation analysis. For example, did the authors use multiple testing corrections? This would ensure transparency and help assess the robustness of the findings.
4. The authors showed the phenotypes of gut microbiota and inflammation, if the authors could add some mechanisms such as reversed experiments or signaling pathways, it will be better.

Minor:

1. Please provide different bacterial CFU, instead of OD values in line 469-470.
2. Please distinguish between 16S rRNA and 16S rDNA.
3. Please provide the ID of the 16S sequencing data.
4. AOM/DSS treated mice ($n=3$) in Fig 1 is little, would lead to unreliable statistics.
5. There are lots of typos and wrong descriptions in the manuscript such as "Feeding stools" in line 95.

Reviewer #3 (Public repository details (Required)):

the authors should deposit their 16S rRNA sequencing data in appropriate public database according to the publisher policy.

Reviewer #3 (Comments for the Author):

Please see the review attached.

General comments:

The authors utilized a traditional CRC mouse model to identify bacteria species that contributed to (*Eubacterium dolichum*, E.d) or prevent CRC occurrence (*Fibrobacter succinogenes*, F.s and *Ruminococcus flavefaciens*, R.f). They further provided convincing evidence that Ed treatment activated the NF- κ B pathway and promoted accumulation of MDSCs and macrophages, thus contributing to the CRC development. In addition, the authors also examined the preventive effect of F.s and R.f on CRC development and immune hemostasis, although the evidence from various aspects were inconsistent. Overall, the manuscript was well written and the results extended our knowledge of casual effect of single bacterial species on CRC susceptibility and possible mechanism of immune regulation involved. However, I believe that there are still some issues with regard to the experiment and interpretation, which I highlight below.

Major Comments:

1. In the implantation experiment with E.d, F.s or R.f, these authors used model strains from ATCC, it is well-known that even the same species from different origins could harbor distinct functions, so it is important to show that strains isolated from their AOM/DSS act similarly as the ATCC strains, at least by in vitro experiments.
2. The authors showed that the gut microbiome in AOM/DSS mouse was altered in figure 2, but how was it affected upon the administration of E.d, F.s and R.f? Are these data available and shall be discussed?
3. I am quite surprised that the author did not mention/discuss the antibiotic cocktail treatment either in the result or discussion section, since this treatment could incur several implications: a) The weak phenotype of Ed or Fs could be due to the antibiotic that eliminate the coordination of other bacteria; b) it would be helpful if the authors show F.s and R.f could colonize in the the gut of AOM/DSS mice.
4. In lines 311-312, the authors claimed R.f had an anti-inflammatory function, yet the only piece of evidence came from modest yet significant increase in CD4+ T cells in the colon. Furthermore, they did not observe significant increase in the spleen, blood or neither in CD8+ T cells. The authors should address or discuss these inconsistencies. What about the levels of anti-inflammatory factors such as IL1, IL6 and TNF-alpha in the tissues mentioned above?
5. I encourage the authors shall discuss further how these three bacterial species regulate inflammation and elicit immune response, although the mechanism in detail may be outside the scope of this manuscript.

Response to reviewer comments

Reviewer #1 (Comments for the Author):

"This study explores the role of gut microbiota in colorectal cancer (CRC) development using an AOM/DSS-induced mouse model. The authors identified Eubacterium dolichum (E.d.) as a CRC-promoting bacterium and Ruminococcus flavefaciens (R.f.) and Fibrobacter succinogenes (F.s.) as potential protective bacteria. While the findings are interesting and contribute to the growing understanding of microbiota in CRC, the study has several limitations that need to be addressed to strengthen its conclusions and translational relevance."

We appreciate the reviewer's positive comments and constructive suggestions on our work.

"1. The study relies on a small number of mice (n = 3 per group for microbiota sequencing), which raises concerns about statistical robustness."

We sincerely appreciate the reviewer's insightful comment regarding the sample size in initial our microbiota sequencing experiments. We acknowledge that the sample size (n=3 per group) is relatively small and may raise concerns about statistical power. However, this experimental design was based on the following considerations:

Since we always observed one or two mice with few colon lumps or even free from tumorigenesis in different AOM/DSS treatment experiments, we intended to exacerbate the induced condition by an elevated DSS concentration to see whether there were still mice free from tumorigenesis in this exacerbated condition, precluding the possibility of inefficient chronic gut damages to be induced. However, the mortality simultaneously increased, albeit not surprisingly, compared to the typical AOM/DSS treatment condition. Therefore, it's virtually impractical to obtain a large number of tumor free animals at the end of our AOM/DSS treatment scheme as we wished, since a large number of experimental animals will be sacrificed unnecessarily during this treatment scheme. Based on our previous experience, approximately 10% mice will be free from tumorigenesis, thus we treated 44 mice with A/D and expected 4~5 mice without tumor formation. However, probably due to the exacerbated induced condition, more mice died before the end of experiment. Eventually, we only had 3 mice without tumor formation.

As far as we are concerned, there is never a "perfect" scheme for tumor modeling using experimental animals such as mice. A fine balance has to be stroke to obtain sufficient number of animals for comparative explorations, while excessive animal usage should be avoided at all cost. We believed a fine balance had been roughly reached in our investigation, given the comparative

initial discovery we made with n=3 each animal group held strong, especially through sufficient and extensive validations by further well controlled, specific bacteria replantation experiments with sufficient large numbers of animals in each group. As we mentioned in the experimental method in the manuscript, we believe our experimental design is thoughtfully devised and our conclusions are uniformly supported by extensive, well controlled investigations well beyond the initial “n=3” pilot discovery.

“2. Do the microbiota identified from the mouse study also apply to human colorectal cancer?”

We thank the reviewer for raising this critical question. While our current study focuses on elucidating the mechanistic role of microbiota influence in a well-controlled murine model, we fully agree that validating these results in human cohorts would significantly further the clinical implications of our study. However, characterizing human CRC promoting or inhibitory microbiota is well beyond the scope of our current study. Therefore, we would like to further our results in a future, well designed, multicenter based, large human cohort with carefully enrolled controls such as chronic IBD patients. Sincere this required serious and rigorous study design as well longitude investigation, we will address this point in full with an independent investigation in our future work.

“3. Did the authors use a germ-free mouse model to assess the impact of specific bacteria on tumor development and progression?”

We thank the reviewer for this insightful question regarding the use of germ-free mouse models in our study. While we acknowledge that germ-free models are a powerful tool for studying the role of specific bacteria in tumor development, we did not employ this approach for the following reasons:

Technical Limitations of Germ-Free Models in AOM/DSS Studies:

As the result in previous studies (Zhan, Chen et al. 2013) as well as our investigations, germ-free mice subjected to the AOM/DSS model often exhibit significantly reduced or even absence of tumor formation. This is likely due to the critical role of the gut microbiota in modulating inflammation and thereafter tumorigenesis. Given that our study aimed to investigate the functional roles of specific bacterial strains in colorectal cancer development, the absence of tumorigenesis in germ-free mice would have hindered our ability to draw meaningful conclusions. To address the confounding effects of a diverse gut microbiota while maintaining tumorigenic potential, we opted for antibiotic pretreatment to deplete the gut microbiota background significantly. Importantly, we observed that antibiotic treatment alone led to a reduction in both

tumor size and number, further supporting the notion that the gut microbiota plays a crucial role in AOM/DSS-induced tumorigenesis. Nevertheless, this approach provided a sensitized background allowing us to assess the impact of specific bacterial strains (E.d, F.s, and R.f) in a controlled yet tumor-permissive environment. In addition, the role of any individual bacteria species in the gut is likely meaningful mainly in the context of a complex microbiota, Therefore, a germ-free model is most useful in suggesting a role of microbiota as a whole, rather than a particular strain during our physiological process.

“4. The study suggests NF-κB activation as a key pathway, but mechanistic validation is minimal. Transcriptomic or cytokine profiling in colonic tissues could provide stronger mechanistic support.”

We think this is an excellent suggestion and we have performed qPCR to further confirmed the changes of the relevant immune cells and the inflammation of the intestinal tissue with classical inflammatory indicators (IL-1β, IL-2, IL-6, TNF-α and IFN-γ). The relative results are shown in additional Figure 4A (new vision) and additional Figure S5. Our new results confirmed the role of cytokines and inflammatory pathways in the tumorigenesis process.

New Result:

In addition, we found that *E.d* administration increased the expression of the pro-inflammatory cytokines including IL-1β, IL-2, IL-6, TNF-α and IFN-γ in the animals regardless AOM/DSS treatment, though some of which failed to reach the statistical significance (Figure 4A).

Figure legend:

Figure 4. *E.d* aggravated intestinal inflammation and activated the NF-κB pathway.

(A) After the mice were sacrificed, the intestinal tissues were collected and homogenized. And

qPCR was performed to detect the relative expression levels of several pro-inflammatory cytokines (IL-1 β , IL-2, IL-6, TNF- α and IFN- γ).

New Result:

In addition, we found R.f administration reduced the expression of the pro-inflammatory cytokines including IL-1 β ($p = 0.0184$), IL-6 ($p = 0.0566$), TNF- α ($p = 0.0167$) and IFN- γ ($p = 0.0850$) in the animals undergoing AOM/DSS treatment (Figure S5A). The pro-inflammatory cytokine (IL-1 β , IL-2, IL-6, TNF- α , IFN- γ) in intestinal tissues did not change significantly with the administration of F.s (Figure S5B).

Figure legend:

Figure S5 Relative abundances of pro-inflammatory marker expression in mouse colon after AOM/DSS induction and R.f/F.s administration.

After the mice were sacrificed, the intestinal tissues were collected and homogenized. And qPCR was performed to detect the relative expression levels of several pro-inflammatory cytokines

(IL-1 β , IL-2, IL-6, TNF- α and IFN- γ) in mice treated by *R.f* (A) and *F.s* (B).

Reviewer #2 (Public repository details (Required)):

“Please provide the ID of 16S rRNA gene sequencing data in a public repository.”

We have deposited our raw data as the reviewer requested:

Data Availability Statement

The microbiome sequencing datasets for this study can be found in the Sequence Read Archive (SRA) database at NCBI under BioProject ID PRJNA799170.

Reviewer #2 (Comments for the Author):

“The manuscript investigated the gut microbiota in AOM/DSS treated mice with and without CRC formation. The results revealed that Ruminococcus flavefaciens and Fibrobacter succinogenes were significantly enriched in CRC-free mice, whereas the presence of Eubacterium dolichum was dramatically reduced. Bacterial intragastric administration experiments demonstrating protective roles of Ruminococcus flavefaciens and Fibrobacter succinogenes, and detrimental role of Eubacterium dolichum in CRC initiation. This paper provides information about effects of gut commensal bacteria on CRC development and it is helpful for the readers on this area. However, there are limited novelty and less mechanism in this study. In addition, some questions in this manuscript should be addressed.”

“Major:

1.16S analysis is not informative enough as to understanding the mechanistic details. None of the results from correlation analysis is confirmed by experiments.”

We appreciate the reviewer’s constructive comments.

We distinguished the relative abundance of *E.d*, *R.f* and *F.s* in mice feces between control mice and AOM/DSS induced mice by qPCR (Figure S1A-C). And we indeed found more *E.d*, less *R.f* and *F.s* in AOM/DSS induced mice compared to the control mice.

Figure legend:

Figure S1. Relative abundance of (A) *Ruminococcus flavefaciens*, (B) *Fibrobacter succinogenes*, and (C) *Eubacterium dolichum* in fecal samples from mice between NC group and AOM/DSS treated- group by qPCR.

“2. Antibiotics like metronidazole are difficult to dissolve in water, and How is the antibiotic cocktail prepared? Antibiotics have a very strong smell and caused mice to dislike drinking water.

Why not use intragastric treatment?”

We appreciate the reviewer’s concerns regarding the preparation and administration of the antibiotic cocktail. Below, we provide a detailed response to address these points:

The antibiotics used in our study, including metronidazole, were prepared at concentrations of up to 1 g/L (0.5 g/L for vancomycin) in drinking water. Under these conditions, we were able to achieve a clear and homogeneous solution. To ensure the stability and efficacy of the antibiotic cocktail, we replaced the drinking water every three days, which maintained the clarity and potency of the solution throughout the treatment period. This approach is consistent with established protocols in the field (Zackular, Baxter et al. 2013).

While we acknowledge that antibiotics can have a strong smell and may affect the palatability of drinking water, we did not observe significant aversion or reduced water consumption in our mice during the treatment period. This suggests that the concentration and preparation method used were well-tolerated by the animals.

We chose oral administration via drinking water rather than intragastric gavage the main reason of mild and sustained depletion: Oral administration allows for a gradual and sustained depletion of the gut microbiota, which we believe is more physiologically relevant and less disruptive than the acute, high-dose exposure associated with intragastric treatment. This approach minimizes the risk of completely obliterating the gut microbiota, which could lead to unintended systemic effects or excessive stress on the animals.

"3.While the manuscript presents many statistical comparisons (e.g., between NC, A/D-T, and A/D-NT groups), it would be helpful to provide more details on the statistical methods used, especially for complex analyses like PCoA, OPLS-DA, and Spearman's correlation analysis. For example, did the authors use multiple testing corrections? This would ensure transparency and help assess the robustness of the findings."

We thank the reviewer for raising this insightful suggestion. We have now included more detailed description on the statistical methods employed in our study in the revised manuscript as suggested.

"4.The authors showed the phenotypes of gut microbiota and inflammation, if the authors could add some mechanisms such as reversed experiments or signaling pathways, it will be better."

This is obviously an important part that all of 3 reviewers have mentioned. As above, we have performed qPCR to further confirmed the changes of the relevant immune cells and the inflammation of the intestinal tissue with classical inflammatory indicators (IL-1 β , IL-2, IL-6, TNF- α and IFN- γ). The relative results are shown in additional Figure 4A (new vision) and additional Figure S5. Our new results confirmed the role of cytokines and inflammatory pathways in the tumorigenesis process.

Minor:

- 1.Please provide different bacterial CFU, instead of OD values in line 469-470.*
- 2.Please distinguish between 16S rRNA and 16S rDNA.*
- 3.Please provide the ID of the 16S sequencing data.*
- 4.AOM/DSS treated mice (n=3) in Fig 1 is little, would lead to unreliable statistics.*
- 5.There are lots of typos and wrong descriptions in the manuscript such as "Feeding stools" in line 95.*

We sincerely thank the reviewer for their careful reading of our manuscript and for identifying several minor issues that needed clarification or correction. We have carefully addressed all of these points in the revised manuscript. All changes have been highlighted in the revised manuscript for the reviewer's convenience. We believe these revisions have improved the clarity and precision of our work, and we are grateful for the reviewer's valuable feedback.

Reviewer #3 (Public repository details (Required)):

the authors should deposit their 16S rRNA sequencing data in appropriate public database according to the publisher policy.

We have deposited our raw data as the reviewer requested:

Data Availability Statement

The microbiome sequencing datasets for this study can be found in the Sequence Read Archive (SRA) database at NCBI under BioProject ID PRJNA799170.

Reviewer #3 (Comments for the Author):

“General comments:

The authors utilized a traditional CRC mouse model to identify bacteria species that contributed to (Eubacterium dolichum, E.d) or prevent CRC occurrence (Fibrobacter succinogenes, F.s and Ruminococcus flavefaciens, R.f). They further provided convincing evidence that Ed treatment activated the NF- κ B pathway and promoted accumulation of MDSCs and macrophages, thus contributing to the CRC development. In addition, the authors also examined the preventive effect of F.s and R.f on CRC development and immune hemostasis, although the evidence from various aspects were inconsistent. Overall, the manuscript was well written and the results extended our knowledge of casual effect of single bacterial species on CRC susceptibility and possible mechanism of immune regulation involved. However, I believe that there are still some issues with regard to the experiment and interpretation, which I highlight below.”

We appreciate the reviewer’s positive comments and constructive suggestions on our work.

“Major Comments:

1. In the implantation experiment with E.d, F.s or R.f, these authors used model strains from ATCC, it is well-known that even the same species from different origins could harbor distinct functions, so it is important to show that strains isolated from their AOM/DSS act similarly as the ATCC strains, at least by in vitro experiments.”

We sincerely appreciate the reviewer’s insightful comment regarding the functional consistency between the ATCC strains and potential isolates from the AOM/DSS model. We agree that strains of the same species from different origins may exhibit distinct functional properties. However, we

would like to clarify that this study represents a preliminary investigation into the functional roles of these bacterial strains (E.d, F.s, and R.f) in colorectal cancer development.

Our initial identification of these strains was based on sequencing data and bioinformatic analysis, followed by functional validation using ATCC strains as a starting point. While we acknowledge that isolating and characterizing strains directly from the AOM/DSS model would provide additional insights, this approach presents significant technical challenges, including the complexity of isolating specific strains from a highly dynamic and diverse gut microbiota environment. Furthermore, there might be a diverse range of strains from the animals themselves, which complicates tremendously results interpretation from a few isolates. Therefore, we employed the “generic” ATCC strains to stratify the “within species variations”, with the results consistent with the hypothesis that those bacteria as a generic population are acting in the direction as our initial finding.

Given the exploratory nature of this work, we acknowledge there could be important strain specific contributions to the overall tumorigenesis process and would like to address this important issue with a detailed future study.

“2. The authors showed that the gut microbiome in AOM/DSS mouse was altered in figure 2, but how was it affected upon the administration of E.d, F.s and R.f? Are these data available and shall be discussed?”

Our study primarily focused on investigating tumor growth and immune-related mechanisms, and thus, we did not monitor changes in the gut microbiome after bacterial inoculation.

This decision was based on two main considerations. First, the administration of antibiotics prior to AOM/DSS modeling significantly altered the composition of the gut microbiota, effectively disrupting its native state (Zackular, Baxter et al. 2013). Consequently, analyzing post-treatment microbiota changes may not provide meaningful insights under these conditions. Second, the primary objective of this study was to validate and explore the effects of the three identified bacterial strains on the initiation and progression of colorectal cancer. Investigating potential interactions or influences with other bacterial species falls beyond the scope of this work.

“3. I am quite surprised that the author did not mention/discuss the antibiotic cocktail treatment either in the result or discussion section, since this treatment could incur several implications: a) The weak phenotype of Ed or Fs could be due to the antibiotic that eliminate the coordination of other bacteria; b) it would be helpful if the authors show Fs and R.f could colonize in the the gut

of AOM/DSS mice.”

We thank the reviewer for raising this important suggestion. We have now included more discussion on the potential role of cocktail alone in limiting epithelial over-proliferation and the significant reduction in tumor size and numbers by treating mice with antibiotics (as seen in Zackular, Baxter et al. 2013). However, we believe our phenotype of Ed or Fs are significant since we used cocktail treatment alone as our control group in both sets of experiments. Therefore, we believe the difference between Ed or Fs administration groups with their antibiotics treatment control group indeed reflect the functional contributions of those bacteria introduced after microbiota clearance.

“4. In lines 311-312, the authors claimed R.f had an anti-inflammatory function, yet the only piece of evidence came from modest yet significant increase in CD4+ T cells in the colon. Furthermore, they did not observe significant increase in the spleen, blood or neither in CD8+ T cells. The authors should address or discuss these inconsistencies. What about the levels of anti-inflammatory factors such as IL1, IL6 and TNF-alpha in the tissues mentioned above?”

As above, we have performed qPCR to further confirmed the changes of the relevant immune cells and the inflammation of the intestinal tissue with classical inflammatory indicators (IL-1 β , IL-2, IL-6, TNF- α and IFN- γ). The relative results are shown in additional Figure 4A (new vision) and additional Figure S5. Our new results confirmed the role of cytokines and inflammatory pathways in the tumorigenesis process.

“5. I encourage the authors shall discuss further how these three bacterial species regulate inflammation and elicit immune response, although the mechanism in detail may be outside the scope of this manuscript.”

We have included additional discussions as the reviewer suggested.

REFERENCE

- Zackular, J. P., N. T. Baxter, K. D. Iverson, W. D. Sadler, J. F. Petrosino, G. Y. Chen and P. D. Schloss (2013). "The gut microbiome modulates colon tumorigenesis." *mBio* **4**(6): e00692-00613.
- Zhan, Y., P. J. Chen, W. D. Sadler, F. Wang, S. Poe, G. Núñez, K. A. Eaton and G. Y. Chen (2013). "Gut microbiota protects against gastrointestinal tumorigenesis caused by epithelial injury."

Cancer Res 73(24): 7199-7210.

Re: Spectrum02792-24R1 (Gut Commensal Bacteria Influence Colorectal Cancer Development by Modulating Immune Response in AOM/DSS-treated Mice)

Dear Dr. Danlei ZHOU:

Thank you for addressing the Reviewers' comments. I would hereby like to congratulate you on the acceptance of your manuscript for publication in Spectrum!

Your manuscript has been accepted, and I am forwarding it to the ASM production staff for publication. Your paper will first be checked to make sure all elements meet the technical requirements. ASM staff will contact you if anything needs to be revised before copyediting and production can begin. Otherwise, you will be notified when your proofs are ready to be viewed.

Sincerely,
Jan Claesen
Editor
Microbiology Spectrum